# 6S-Like scr3559 RNA Affects Development and Antibiotic Production in *Streptomyces coelicolor*

**DOI:** 10.3390/microorganisms9102004

**Published:** 2021-09-22

**Authors:** Jan Bobek, Adéla Mikulová, Dita Šetinová, Marie Elliot, Matouš Čihák

**Affiliations:** 1Institute of Immunology and Microbiology, 1st Faculty of Medicine, Charles University, Studničkova 7, 12800 Prague, Czech Republic; Dita.Setinova@lf1.cuni.cz (D.Š.); matous.cihak@lf1.cuni.cz (M.Č.); 2Faculty of Science, Jan Evangelista Purkyně University in Ústí nad Labem, České mládeže 8, 40096 Ústí nad Labem, Czech Republic; adimiky@gmail.com; 3Department of Biology, McMaster University, Hamilton, ON L8S 4K1, Canada; melliot@mcmaster.ca

**Keywords:** small RNA, 6S RNA, *Streptomyces*, antibiotics, secondary metabolism

## Abstract

Regulatory RNAs control a number of physiological processes in bacterial cells. Here we report on a 6S-like RNA transcript (*scr3559*) that affects both development and antibiotic production in *Streptomyces coelicolor*. Its expression is enhanced during the transition to stationary phase. Strains that over-expressed the *scr3559* gene region exhibited a shortened exponential growth phase in comparison with a control strain; accelerated aerial mycelium formation and spore maturation; alongside an elevated production of actinorhodin and undecylprodigiosin. These observations were supported by LC-MS analyses of other produced metabolites, including: germicidins, desferrioxamines, and coelimycin. A subsequent microarray differential analysis revealed increased expression of genes associated with the described morphological and physiological changes. Structural and functional similarities between the *scr3559* transcript and 6S RNA, and its possible employment in regulating secondary metabolite production are discussed.

## 1. Introduction

Streptomycetes are Gram-positive soil-dwelling bacteria. The soil niche represents a harsh living environment where the natural microbial inhabitants have evolved various nutrient-acquiring and life-defending strategies. *Streptomyces* have a complex developmental life cycle that begins with germinating spores. They then form branching vegetative hyphae that differentiate into aerial mycelium and spores again [1,2,3,4]. Moreover, the streptomycetes possess extraordinary genetic equipment for sensing extracellular signals and producing various specialized metabolites [5]. Among other applications, these molecules encompass over two-thirds of the clinically useful antibiotics and other compounds of industrial value [6]. The developmental complexity and antibiotic biosynthesis of these organisms require a complex regulatory network, which is responsible for addressing proper responses to changes in environmental conditions [7].

In order to achieve such responses, a number of pleiotropic regulators are involved in the gene expression control of the streptomycetes. So-called regulatory RNAs represent one such regulatory sphere. A single RNA molecule may regulate multiple genes, thus broadly influencing the physiology of the cell [8]. These RNAs act either simply by base-pairing with target nucleotides, or by binding target proteins via their secondary structures. 6S RNA achieves gene regulation using the second strategy and is one of the best studied regulatory RNAs in bacteria [9]. 6S RNA has a direct influence on the transition between developmental stages thanks to its direct association with the sigma70-RNA polymerase holoenzymes (RNAP-HrdB in *Streptomyces*). The formation of such complexes interferes with exponential phase-specific transcription [10,11,12]. Due to its function, the promoter-like secondary structure of 6S RNA is highly conserved across the bacterial kingdom. This high degree of conservation has enabled the identification of more than 100 potential 6S RNA homologues in diverse eubacterial species, using computational searches [13]. Notably, however, *Streptomyces* were absent among those species in which 6S RNA was identified.

We had developed a bioinformatic tool that detected two streptomycete 6S-like transcripts whose secondary structures exhibited convincing similarity with those of 6S RNAs found in other bacteria [14,15]. Subsequently, it was shown that a null mutation of one of these genes (*scr3559*), affected actinorhodin production in *S. coelicolor* [16]. To analyze the effect of the scr3559 RNA on cell morphology and physiology, we constructed a strain that over-expresses *scr3559*, with its gene placed under the control of a thiostrepton-inducible promoter. Interestingly, the levels of the scr3559 RNA were consistent with a strong effect on the exponential growth of *S. coelicolor*, as well as on its secondary metabolite production. This suggests that the scr3559 RNA functions as an important developmental regulator.

## 2. Materials and Methods

### 2.1. Strains, Media and Conditions of Cultivation

*Streptomyces coelicolor* M145 was used as a model wild type strain, as well as the parental strain used to construct a strain that over-expresses the *scr3559* gene. The resulting *S. coelicolor* strains, *Escherichia coli* strains, and all plasmids/cosmids used in this study are summarized in Table 1.

*E. coli* strains were grown at 37 °C on solid agar and in LB liquid medium [17]. *Streptomyces* strains (10^8^ spores) were grown at 30 °C on the solid agar media soya flour-mannitol (MS), supplemented minimal (SMMS) [18], and Oxoid Nutrient Agar (ONA, OXOID CZ s.r.o, Thermo Fisher Scientific, Brno, Czech Republic) or in liquid rich with glucose, unless otherwise stated, media (R3) [19] or minimal NMMP [18].

### 2.2. Construction of High-Copy-Number Vector Overexpression Strains

The over-expression strain used in this study was constructed using the actinomycete high-copy-number shuttle vector pCJW93 (Table 1). The *scr3559* gene from *S. coelicolor* was PCR amplified from cosmid StH5 (Table 1) using primers 6SO/E1 (GTGCGCGCCTGTCCGCGCTTG) and 6SO/E2 (GCCGATCGGTCGTCGGTTCGC), and the product was inserted into pIJ2925 (Table 1) using a rapid DNA ligation kit (Roche, Merck Life Science spol. s r.o., Prague, Czech Republic). The construct was transformed into DH5α *E. coli* cells (Table 1). The *scr3559*-containing fragment was cut out from the plasmid with BglII and subcloned into the BamHI site of the over-expression vector pCJW93, downstream from the thiostrepton-inducible (*tipA*) promoter. The plasmid was passaged through the non-methylating *E. coli* strain ET12567/pUZ8002 (Table 1) before being introduced into *S. coelicolor* M145 by conjugation [18], resulting in the construction of the C6S over-expression strain. A control strain, C0, containing the pCJW93 vector lacking any insert was constructed simultaneously.

### 2.3. Mapping of 5′ Ends

The 5′ rapid amplification of cDNA ends (5′ RACE) was performed using the FirstChoice^®^ RLM RACE kit (Invitrogen, Thermo Fisher Scientific, Brno, Czech Republic) with minor modifications, as described in [24]. Total RNA was extracted from a plasmid-free strain of *S. coelicolor* that had been cultivated for 72 h in NMMP liquid medium.

RNA quality was assessed by determining the RNA-integrity number (RIN) and only those samples with a RIN higher than 7 were further analyzed. Ten micrograms of RNA were treated with tobacco acid pyrophosphatase (TAP) before being ligated to a 5′ RACE adapter using T4 RNA ligase. The obtained product was used as template for reverse transcription using 50 µM of random decamers and M-MLV reverse transcriptase.

The resulting cDNA served as a template for 5′ end PCR amplification using an outer adaptor-specific oligonucleotide and an oligonucleotide complementary to the scr3559 RNA sequence (CGCTTACTCGGGACCGGT). The PCR product was used as a template for a second PCR reaction using an inner adaptor-specific oligonucleotide and an inner ‘nested’ scr3559 RNA-complementary primer (6SR; ACCGGTATACAAAGGACTCAACGG). The second PCR product was separated on an agarose gel and eluted using the QIAquick-Gel Extraction kit (Qiagen, GeneTiCA s.r.o., Prague, Czech Republic). The purified products were cloned into the TOPO^®^ vector using TOPO TA Cloning^®^ (Invitrogen) and transformed into TOP10F competent cells (Invitrogen). Plasmid DNA was extracted from cells containing cDNA inserts using the QIAprep^®^ Miniprep kit (Qiagen), and the cloned inserts were sequenced using M13 primers.

### 2.4. Differential Expression Analyses (Northern Blot)

At indicated time points of cultivation in the NMMP liquid medium (with thiostrepton supplemented at 50 µg/mL), cells were harvested and RNA was isolated as described in [25]. UV spectroscopy and agarose gel electrophoresis were used to assess the quantity and quality of total RNA samples. Total RNA samples were separated on 6% denaturing polyacrylamide gels and were transferred to nylon membranes (BioRad s.r.o., Prague, Czech Republic) using a Trans-Blot semi-dry transfer cell (BioRad) (25 V, 4 °C overnight). The membrane was then UV-cross linked before being hybridized with the 5′-end-labeled oligonucleotide 6SR (ACCGGTAGTACAAAGGACTCAACGG), which corresponded to an internal segment of the scr3559 RNA. The hybridization was performed overnight at 42 °C in ULTRA hybridization buffer (Invitrogen). The detection and quantification of signals were conducted using a phosphorimager (BioRad).

### 2.5. Secondary Metabolite Production Assays

To quantify the actinorhodin and undecylprodigiosin levels in the C6S strain relative to the C0 strain, 10^8^ spores of each were inoculated into NMMP liquid medium (with thiostrepton supplemented at 50 µg/mL). Cultures were shaken at 30 °C and samples were taken after 48, 72, and 144 h (unless otherwise stated). Actinorhodin and undecylprodigiosin levels were then quantified as described [26] and amounts were normalized per cell concentration.

### 2.6. Liquid Chromatography-Mass Spectrometry (LC-MS) Assay

Both C0 and C6S strains were cultivated for 48 h in R3 medium containing thiostrepton (50 µg/mL). Secondary metabolites were extracted from the culture supernatants using solid phase extraction [27,28]. LC-MS analyses were performed on the Acquity UPLC system with 2996 PDA detection system (194–600 nm) connected to LCT premier XE time-of-flight mass spectrometer (Waters Gesellschaft m.b.H., Prague, Czech Republic). Five microliters of the sample were loaded onto the Acquity UPLC BEH C18 LC column (50 mm × 2.1 mm I.D., particle size 1.7 μm, Waters) at 40 °C and eluted with a two-component mobile phase, A and B, consisting of 0.1% formic acid and acetonitrile, respectively, at the flow rate of 0.4 mL min^−1^.

The analyses were performed under a linear gradient program (min/%B) 0/5; 1.5/5; 15/70; 18/99 followed by a 1.0-min column clean-up (99% B) and 1.5-min equilibration (5% B). The mass spectrometer was operated in the positive “W” mode with capillary voltage set at +2800 V, cone voltage +40 V, desolvation gas temperature, 350 °C; ion source block temperature, 120 °C; cone gas flow, 50 L h^−1^; desolvation gas flow, 800 L h^−1^; scan time of 0.15 s; inter-scan delay of 0.01 s. The mass accuracy was kept below 6 ppm using lock spray technology, with leucine enkephalin being used as the reference compound (2 ng μL^−1^, 5 μL min^−1^). MS chromatograms were extracted for [M + H]^+^ ions with the tolerance window of 0.03 Da, smoothed with a mean smoothing method (window size; 4 scans, number of smooths, 2). The data were processed by MassLynx V4.1 (Waters).

### 2.7. Microarrays

Cultivation of the C6S and control C0 strains was performed in duplicate. 10^8^ spores were inoculated into NMMP liquid medium (with thiostrepton at 50 µg/mL) and shaken at 30 °C. Cells were harvested after 72 and 144 h and the quality of the isolated RNA samples was determined by their RIN. Only those samples whose RIN was >7 were sent to the Oxford Gene Technology Company (OGT, Oxford, United Kingdom), where microarray analysis was performed as follows: the RNA samples were used as template for reverse transcription using random hexamer primers and Cy3/Cy5 dyes (swapped in the duplicates); the labeled cDNAs were hybridized on microarray slides containing 4 × 44,000 spotted oligonucleotides, mapped preferentially to the ORFs of the *S. coelicolor* genome. The differential expression was estimated using the binary logarithm of the ratio of the expression level in the test C6S sample to the level of expression in the control C0 sample.

## 3. Results

### 3.1. Size and Gene Position of the scr3559 Transcript

The scr3559 RNA (also called Sc2 in [15] or 6S RNA in [16]) is a 192 nt long transcript that folds into a 6S-like secondary structure [16]. Its gene is located between *SCO3558* and *SCO3559*. It is noteworthy that it is oriented downstream of *SCO3559*, as illustrated here in Figure 1 (minus strand) and is in accordance with the RNA sequence shown in both the above-mentioned publications, although the gene is mistakenly depicted in the opposite direction in Mikulík et al. [16], where it is dubbed *ssrS*.

To precisely map the *scr3559* transcription start site, a RACE experiment was performed using internal nested primers complementary to the scr3559 RNA. Two RT-PCR products were detected and sequenced (Figure 2). These corresponded to a nascent (pre-scr3559 RNA) and a processed transcript (scr3559 RNA) of sizes 260 and 192 bases, respectively. The transcription start site is thus located at genome position 3934888, whereas the processed 6S-like form, as presented previously, begins at position 3934820. Based on these results, secondary structures of precursor (pre-scr3559 RNA), processed (scr3559 RNA) and excised transcripts were modeled using the RNAfold algorithm (http://rna.tbi.univie.ac.at/cgi-bin/RNAWebSuite/RNAfold.cgi, accessed on 1 September 2021) [29] (Figure 2b).

### 3.2. scr3559 RNA Expression Profiles (Northern Blotting)

scr3559 RNA levels were monitored throughout development for both the control C0 strain and the mutant C6S strain. Cells were cultivated in liquid NMMP medium and harvested after 12 h (early exponential phase of growth), 48 h (late exponential phase of growth) and 72 h of cultivation (stationary phase). No signal was detected in samples from the early exponential phase (not shown). A signal corresponding with pre-scr3559 RNA (260 nt) was detected only in the older cells of the C0 strain, whereas it is clearly seen in both developmental time points in the C6S over-expression strain (Figure 3). In both strains, the processed scr3559 RNA could only be detected during the exponential to stationary phase transition, when 6S RNA is expected to be active.

### 3.3. Phenotypic Analyses

After cultivation in NMMP medium with the selective antibiotic thiostrepton, the production of actinorhodin and undecylprodigiosin metabolites were measured and normalized per cell concentration. The dry weight of the cells was estimated after they were collected in spin columns (Qiagen), centrifuged and washed with ethanol. Elevated antibiotic production, actinorhodin (blue) and undecylprodigiosin (red) were repeatedly detected in the C6S overexpression strain, compared with the C0 control strain (Figure 4).

When cultivated on solid agar plates containing thiostrepton (R2YE, SMMS, MS), the C6S strains revealed differences in morphology (small colonies, aberrant mycelium formation, less biomass, and more rapid sporulation compared to the C0 control), and exhibited increased production of secondary metabolites (mainly blue actinorhodin) (Figure 5a). This observation was supported by the data collected using electron microscopy to compare the morphology of C6S and C0 control strains grown in the presence of thiostrepton. The pictures clearly show the accelerated development of the aerial mycelium for the C6S strain (Figure 5b).

When cultivated on solid MS plates with apramycin and a gradient of thiostrepton, actinorhodin was visibly over-produced by the C6S strain only in the areas with higher concentrations of thiostrepton. In this area, growth of the C6S strain was slower, its cell density was lower and colonies were tiny and grey, suggesting that vegetative growth may be suppressed, and sporulation may be accelerated.

Growth in the area that did not contain thiostrepton (i.e., the *tipA* promoter is not induced and *scr3559* is not overexpressed) looked similar to that of the C0 control strain, which did not show any phenotypic differences on the same thiostrepton gradient on MS medium. When cultivated on ONA medium with an equivalent thiostrepton gradient, the C6S strain produced visibly more actinorhodin throughout the entire plate. Only in the area with the highest thiostrepton concentrations, actinorhodin was overproduced in both strains. (Figure 6). This supports our assertion that overexpressing *scr3559* (in response to thiostrepton induction) leads to profound phenotypic changes.

### 3.4. LC-MS Analysis of Other Metabolite Production

After both C6S and C0 strains had been cultivated for 48 h in R3 liquid medium, supernatants were subjected to LC-MS analysis. No qualitative changes were observed in the production of secondary metabolites, but quantitative differences were detected. Besides the already described increased production of actinorhodin and undecylprodigiosin, other metabolites were detected in both strains. These include germicidins (A, B), desferrioxamines (B, E, G1), and coelimycin P1.

#### 3.4.1. Germicidins

The LC-MS data qualitatively confirmed the presence of germicidin A (tR = 6.48 min) and germicidin B (tR = 5.91 min). The acquired *m*/*z 197.1193* corresponded to the [M + H]^+^ of germicidin A with a theoretical *m*/*z 197.1178*. Based on a comparison of the area under the curve (AUC), six times higher production of germicidin A was observed in C6S (AUC_C6S_ = 1252) compared with the C0 strain (AUC_C0_ = 206) (Figure 7). Germicidin B was also produced at higher levels in the C6S strain as well, although its levels were not as strikingly enhanced relative to that of germicidin A (AUC_C6S_ = 336 and AUC_C0_ = 90). The acquired *m*/*z 183.1031* corresponded to the [M + H]^+^ of germicidin B with a theoretical *m*/*z 183.1021*.

#### 3.4.2. Desferrioxaomines

Desferrioxamine B was detected at tR = 3.14 min. The acquired *m*/*z 561.3614* corresponded to the [M + H]^+^ of desferrioxamine B with a theoretical *m*/*z 561.3625*. The production of this metabolite during the cultivation in R3 with glucose was higher in C6S (AUC_C6S_ = 173) than in C0 (AUC_C0_ = 110) (Figure 8). A similar trend was seen for the production of desferrioxamine E (AUC_C6S_ = 208 vs. AUC_C0_ = 139) and G1 (AUC_C6S_ = 34 vs. AUC_C0_ = 21). Desferrioxamine E was detected at tR = 4.72 min. The acquired *m*/*z 601.3546* corresponded to the [M + H]^+^ of desferrioxamine E with a theoretical *m*/*z 601.3561*. Desferrioxamine G1 was also detected (tR = 3.28; *m*/*z 619.3663*), although the differences between the C6S and C0 strains were subtle.

#### 3.4.3. Coelimycin P1

We also confirmed the presence of coelimycin P1 (tR = 7.48), an unusual polyketide alkaloid produced by *S. coelicolor* [30]. The acquired *m*/*z 349.1222* corresponded to the [M + H]^+^ of coelimycin P1 with a theoretical *m*/*z 349.1229*. Traces of this substance were detected when culturing both strains in R3 liquid medium with glucose (AUC_C6S_ = 21 vs. AUC_C0_ = 8) (Figure 9). However, when we used glycerol instead of glucose as an energy source in the R3 cultivation medium, the production of coelimycin was apparent only in the C6S strain (AUC_C6S_ = 139).

### 3.5. Microarray Transcriptional Analysis

Microarray analysis was performed on C0 and C6S cells grown in liquid NMMP medium. Since previous expression analysis had shown higher levels of scr3559 RNA at later stages of growth, two stages of cell development were investigated: 72 h (younger cells), corresponding to early stationary phase (transition from exponential growth phase, when antibiotic production typically initiates) and 144 h (older cells), corresponding to late stationary stage. Genes that exhibited differential changes in expression are summarized in Table 2 and discussed below.

## 4. Discussion

The regulatory capabilities of small RNAs have only been appreciated for the last 20 years or so. This lack of awareness is exemplified in work by Nishiyama et al. [31], where the authors created a *Streptomyces azureus* strain that overexpressed the BalA1 region of the chromosome (a 2.5-kb chromosomal DNA fragment). The resulting BalA1 strain formed little aerial hyphae and exhibited a capacity to overproduce secondary metabolites. The authors focused on the fact that the equivalent chromosomal region in the *S. coelicolor* genome contains two protein-coding genes: *SCO3557* and *SCO3558*. They referred to *SCO3558* as encoding a “morphological differentiation-associated protein”. Subsequent work [14,24,32] has revealed, however, that the *SCO3558* gene is flanked by two sRNA-coding genes *scr3558* (encoding 6C RNA) [24,33] and by the scr3559 (investigated here) (Figure 1). Expression of both scr3558 and scr3559 RNAs has been verified. Mutagenesis of the *scr3558* gene had no effect on cell physiology (personal communication [34]), whereas deletion [16] and/or overproduction of the scr3559 RNA (our C6S strain) led to changes in morphological development and secondary metabolite production. The phenotypes in this study are reminiscent of those described for the *S. azureus* BalA1 strain, where the scr3559-equivalent RNA could be responsible for the observed change in cell physiology following overexpression of the sRNA region.

One may speculate that the BalA1 phenotype could be caused by increased expression of other gene product(s) from the BalA1 locus, including SCO3558 or SCO3559. The *SCO3558* gene is a possible paralogue of the *mmfP* gene, which is located in a cluster of genes encoding enzymes responsible for the synthesis of the methylenomycin furans (MMFs) from the SCP1 plasmid of *S. coelicolor* [35]. These MMFs are freely diffusible signaling molecules that control the expression of the plasmid-encoded antibiotic methylenomycin. Nishiyama et al. also demonstrated that the *S. azureus* BalA1 strain induced its neighboring *S. coelicolor* A3(2) to produce the red antibiotic undecylprodigiosin early on in its life cycle. We therefore speculated that the scr3559 RNA in *S. coelicolor* may also contribute to the control of signaling molecule(s). However, we did not see any influence of the C6S strain on the development of the neighboring C0 strain, as was observed in the BalA1 strain. It is not yet clear whether either of the flanking protein-coding genes (*SCO3558* and *SCO3559*), contribute to *Streptomyces* development, as their expression decreased in the C6S strain according to our microarray data.

The results presented here indicate that sRNAs contribute to the control of antibiotic production in *Streptomyces* species. Several pioneering studies on specific small RNAs expressed in *Streptomyces* have been published [36,37,38]. Subsequent bioinformatic approaches have predicted that tens of sRNA transcripts might be expressed during the developmental cycle of *S. coelicolor* [14,24,25,32,39,40]. The scr3559 RNA, whose pleiotropic phenotypic effects are described here, was first detected by Panek [14]. Further study revealed that the transcript’s secondary structure is reminiscent of the well-characterized 6S RNA [15]. Binding assays confirmed that *scr3559* can form a complex with RNAP-HrdB and an analysis of its secondary structure validated these earlier predictions [16].

As we found here, two forms of scr3559 RNA are produced: a longer 260 nt-long molecule referred to here as pre-scr3559, and a shorter 192 nt-long “processed” form. 6S RNA processing has been studied in *E. coli* and other proteobacteria, where two precursor transcripts (transcribed from different promoters) are processed to give a mature 183 nt 6S RNA [41]. RNase E and its paralogue RNase G are required for this processing [41]. A similar situation has been noted in *Bacillus thuringiensis*, where two genes, *ssrSA* and *ssrSB*, are co-transcribed and processed by ribonucleases to give a functional 6S RNA [42]. It would be expected that equivalent RNases may contribute to the processing of the 6S RNA in *S. coelicolor* (Figure 2b); however, this has not yet been explored experimentally. It is also not known whether both precursor and processed-6S RNAs are functional, or whether functionality is confined to the processed form.

Interestingly, a deep sequencing analysis of *S. coelicolor* transcriptome detected a transcript antisense to the scr3559 RNA which is also expressed in the early stationary phase of cell development [32]. We believe that this could be a transcript equivalent to Ms1 RNA found in *Mycobacterium smegmatis* where it is positioned in the same antisense orientation [15,43]. Ms1 RNA accumulates in stationary phase and forms a 6S-like secondary structure [44]. Therefore, Ms1 RNA has been proposed to be the mycobacterial 6S RNA. It was however later demonstrated that Ms1 does not bind to the primary σ^70^-factor-RNAP complex, and instead forms a complex with the RNAP core that is devoid of the sigma factor [44]. This promotes increased sigma-free core RNA polymerase levels, leading to a reservoir of RNAP in stationary phase [45]. Further research is needed in order to elucidate the roles of both the 6S-like scr3559 RNA and its antisense Ms1-like transcript.

6S RNA levels are at their maximum during stationary phase in many bacteria [46]. In *E. coli*, 6S RNA levels increase up to 11 times in stationary phase compared to exponential phase [47], and the transcript effectively sequesters the majority of σ^70^ and upregulates σ^S^ synthesis [46,48,49]. This promotes optimal cell survival, promoting sigma factor switching by the transcriptional machinery and leading to different transcriptional outcomes. This allows the bacteria to better adapt to and survive various stress conditions including nutrient limitation, and prolongs cell viability when compared with 6S RNA-deficient strains [46].

*Bacillus subtilis* is slightly more complex, as it expresses two 6S RNAs: 6S-1 RNA and 6S-2 RNA. Whereas 6S-1 RNA exhibits a similar expression profile to the *E. coli* 6S RNA, 6S-2 RNA’s expression is steady throughout the cell cycle [13,50,51,52]. Interestingly, 6S-1 RNA impacts the timing of sporulation [53]. A strain of *B. thuringiensis* lacking 6S-1 RNA also exhibited a reduced capacity to survive stationary phase and was defective in spore formation [42]. 6S RNA has also been observed to mediate transition into a dormant state in *Coxiella burnetii*, an obligate intracellular pathogen. The cellular level of 6S RNA in this bacterium peaks after developmental differentiation into a spore-like small cell variant [54].

These findings are all consistent with our observations in *Streptomyces*. As we have shown, scr3559 RNA levels increase during the transition into stationary phase, and culminate with onset of antibiotic production. When *scr3559* expression was induced by thiostrepton, the C6S strain was observed to promote rapid entry into stationary phase. On solid media, this resulted in the formation of very small, precociously sporulating colonies (Figure 5).

These observations correlated with our microarray analyses, where we observed increased transcript levels for genes whose products drive aerial hyphae development, septation and sporulation in the C6S strain. Although some streptomycete strains, like *S. coelicolor*, do not develop aerial mycelium and spores in liquid media, transcriptomic data suggest that gene expression in liquid and solid cultures are comparable [55]. A comparative transcriptomic study revealed that 86% of genes are expressed at the same level in both liquid and solid media, including genes involved in sporulation (*bldN*, *sapA*, and *whiG*, among others) [55].

The alternative BldN-sigma factor of RNAP is required for aerial mycelium formation and subsequent sporulation [56]. As our microarray data show, transcript levels of this developmental regulator in the C6S strain exhibited an almost four-fold increase in younger cells (72 h) and an approximate two and a half fold increase in older cultures (144 h) compared with the control strain C0.

Early stages of sporulation are regulated by WhiG, another alternative sigma factor of RNAP [57]. Its regulon members include *whiH*, which encodes a sporulation transcription factor [58,59,60]. WhiH is required for the process of aerial hyphae septation; only a few sporulation septa are formed in loosely coiled aerial hyphae when this GntR-family transcription factor is missing [58]. *whiH* also activates the expression of *geoA*, a gene encoding an enzyme responsible for the synthesis of the earthy volatile compound geosmin [61]. The expression of *whiH* and *whiG* genes was doubled and tripled, respectively, in the older C6S strain. Also upregulated in C6S was the expression of the spore-associated protein SapA-encoding gene, whose synthesis is also connected with the aerial hyphae development [62,63].

Morphological development during stationary phase is accompanied by increased secondary metabolism; this was even more active when high levels of scr3559 RNA were produced. One explanation for this could be through elevated levels of the γ-butyrolactone bacterial hormones. These molecules are responsible for cell signaling, coordinating morphological changes, and triggering the onset of secondary metabolism during stationary phase [64,65]. The synthesis of the *S. coelicolor* butanolide (SCB; an equivalent molecule to A-factor in *S. griseus*) is directed by the γ-butyrolactone synthase ScbA [66]. SCB is sensed by the receptor protein ScbR, whose repression of biosynthetic gene clusters is then relieved. Our microarray data show that in the young C6S strain, expression of *scbA* is upregulated, while expression of *scbR* is reduced. Two other genetic loci, whose expression is known to activate antibiotic production in *S. coelicolor* (the Abe cluster [67] and AbsR [68]), were also upregulated in the younger C6S strain. These activation events may contribute to the increased production of antibiotics.

Other than actinorhodin, the C6S strain produced the undecylprodigiosin (red) antibiotic earlier in its life cycle and in higher concentrations than the control C0 strain. Prodiginines are membrane damaging molecules that are produced inside the substrate mycelia cells in nutrient-deficient conditions. Lacking any self-resistance gene, they may contribute to a programmed cell death process that supports aerial hyphae development [69]. The “red” biosynthetic pathway is activated by the RedD regulator, whose gene expression is upregulated (doubled to tripled) during the development of the C6S strain. This occurs in concert with reduced growth of vegetative hyphae of the C6S strain observed here (Figure 5).

Other upregulated secondary metabolite gene clusters in the C6S strain include *crtEIBV* and *crtYTU* (in older cells), which are associated with the carotenoid isorenieratene synthesis pathway [70] and the Gcs type III polyketide (germicidins) [71], respectively. The expression of the Gcs gene *SCO7221* was tripled in older C6S cells, in accordance with the elevated germicidin levels observed in our LC-MS analysis (Figure 7). Germicidin A inhibits not only spore germination, but also vegetative hyphae elongation [72]. Thus, the germicidins may therefore contribute—possibly in association with the prodiginines—to the sporulation process supported by the scr3559 RNA.

## 5. Conclusions

The phenotypic observations presented here imply that scr3559 RNA plays a central role in the developmental control mechanisms of *Streptomyces* bacteria. The similarities shared by scr3559 RNA and bacterial 6S RNAs [16] may explain its pleiotropic effects on development and secondary metabolism. 6S RNA in *Streptomyces* may function to sequester exponential phase-specific HrdB-RNAP in order to effectively promote entry into stationary phase, leading to the formation of spores. As this development is accompanied by increased activity of secondary metabolism, the 6S RNA role also appears to include an impact on antibiotic production, given the elevated levels observed in the C6S over-expression strain.

## Figures and Tables

**Figure 1 microorganisms-09-02004-f001:**
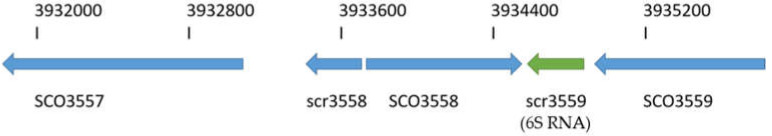
Revised *scr3559* gene orientation, relative to its flanking genes.

**Figure 2 microorganisms-09-02004-f002:**
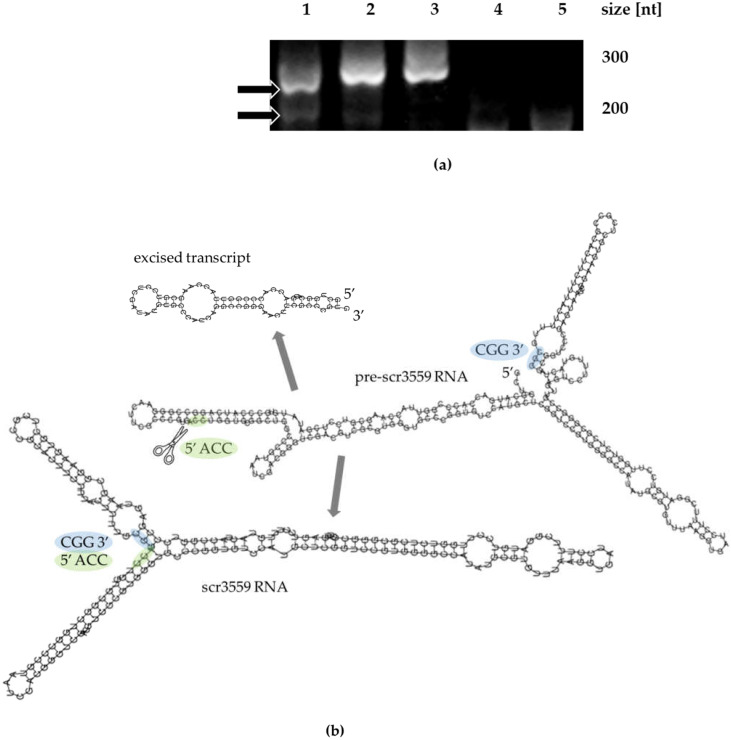
(**a**) 5′ RACE analysis. From the left: lanes 1–3 PCR products from the RACE experiment, taken from 72, 144, and 168-h-old cultures; lanes 4–5 RTase-free and ligase-free negative controls (no signals). The arrows point to those PCR products whose sequences correspond to the nascent scr3559 RNA (260 bases) and to the processed form (192 bases). The molecular sizes of the RACE-generated PCR products roughly corresponded to the sizes of the whole transcripts. (**b**) Secondary structure predictions for the 260 nt pre-scr3559 RNA, a processed 192 nt form and the excised 68 nt transcript. The structures were modeled using RNAfold software with energy parameters scaled to 30 °C. The scissors mark the predicted site of cleavage in the precursor transcript. The green and blue clouds indicate the 5′ and 3′ ends of scr3559 RNA, respectively.

**Figure 3 microorganisms-09-02004-f003:**
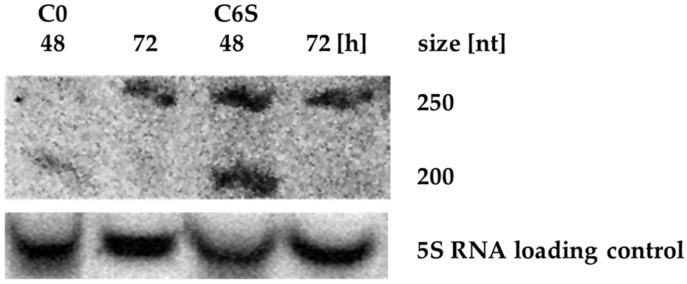
Northern blot analysis of scr3559 RNA levels in C0 and C6S strain. The product at about 250 nt likely corresponds to the pre-scr3559 RNA. The processed scr3559 RNA (192 nt) could be seen only in 48 h-old cells of both C0 and C6S strains.

**Figure 4 microorganisms-09-02004-f004:**
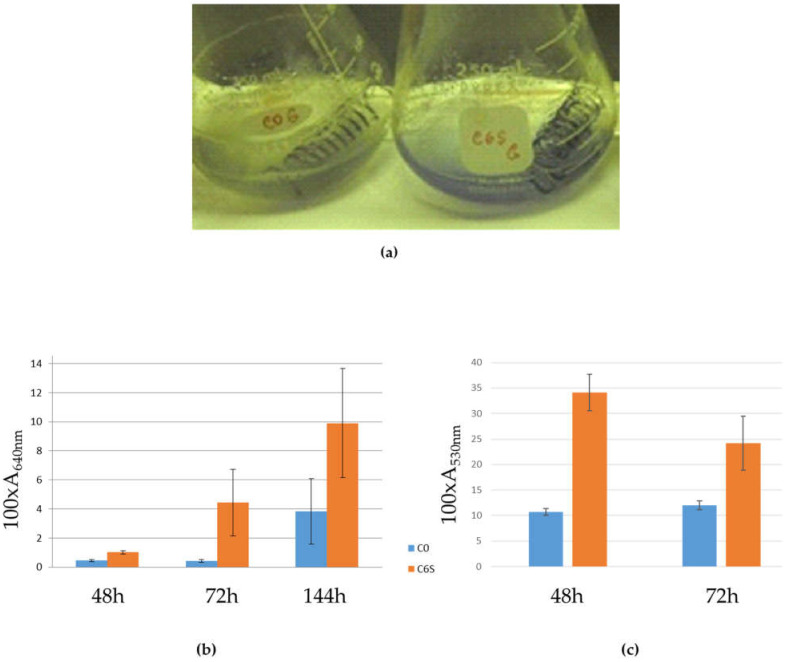
Liquid culture phenotypes: (**a**) The C0 control (left) and C6S (right) strains after 72 h of cultivation in NMMP liquid medium. (**b**) Comparison of actinorhodin production in C0 and C6S strains; the *y*-axis indicates absorbance at A_640_ ×100 and normalized per cell concentration. The error bars represent a confidence interval from at least four independent biological replicates (α = 0.2). (**c**) Comparison of undecylprodigiosin production in C0 and C6S strains; the *y*-axis indicates absorbance at A_530_ × 100 and normalized per cell concentration. The error bars represent a confidence interval from at least three independent biological replicates (α = 0.2).

**Figure 5 microorganisms-09-02004-f005:**
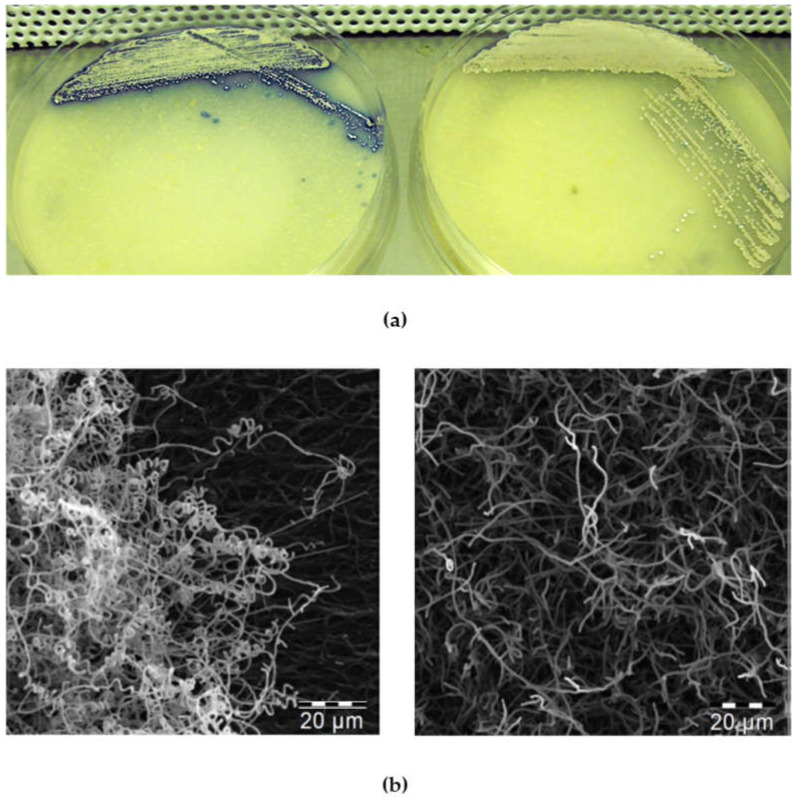
Solid culture phenotypes: (**a**) The C6S (left) and C0 control (right) strains after 120 h of cultivation on MS medium. (**b**) Electron-micrographs of mycelia from the C6S (left) and C0 (right) strains after 120 h of cultivation on solid MS medium.

**Figure 6 microorganisms-09-02004-f006:**
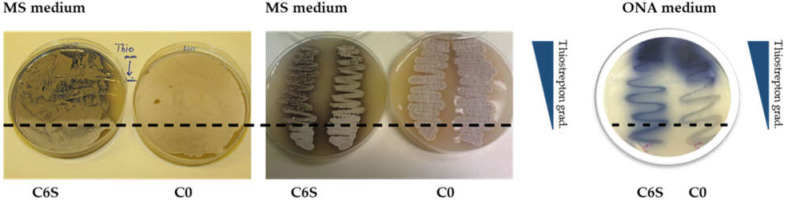
The C6S and C0 strains on MS and ONA medium with a linear concentration gradient of thiostrepton (0–25 μL mL^−1^ respectively to the blue wedge shown to the right of the plates) after 120 h of cultivation. The dashed line indicates where the thiostrepton gradient begins (most concentrated area is at the top of the image).

**Figure 7 microorganisms-09-02004-f007:**
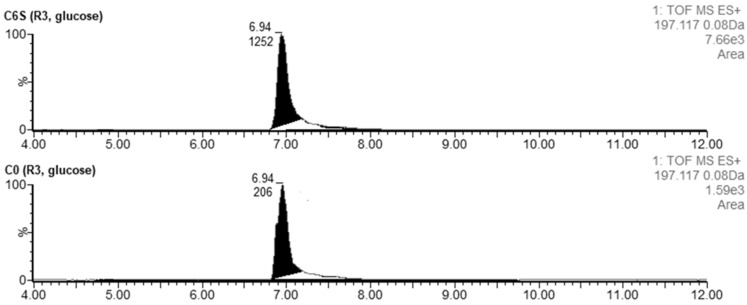
Extracted LC-MS chromatograms of germicidin A produced by *S. coelicolor* C6S and C0 after 48 h of cultivation in R3 medium.

**Figure 8 microorganisms-09-02004-f008:**
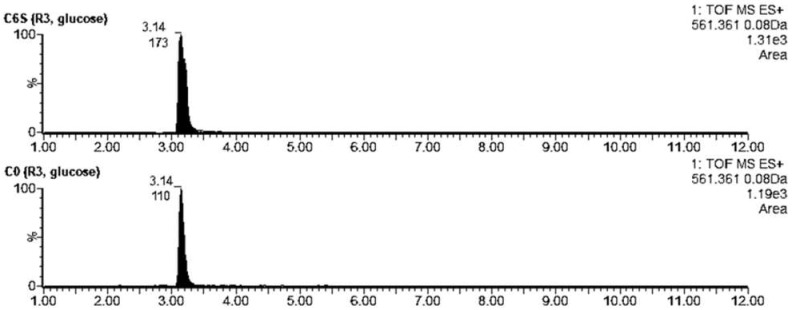
Extracted LC-MS chromatograms of desferrioxamine B produced by *S. coelicolor* C6S and C0 after 48 h of cultivation in R3 medium.

**Figure 9 microorganisms-09-02004-f009:**
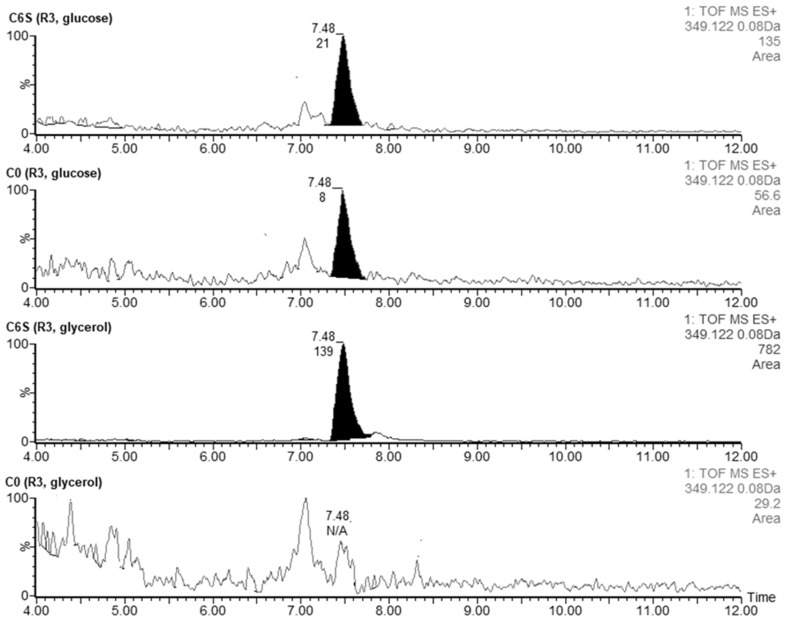
Extracted LC-MS chromatograms of coelimycin P1 produced by *S. coelicolor* C6S and C0 after 48 h of cultivation in R3 medium with glucose or glycerol.

**Table 1 microorganisms-09-02004-t001:** Bacterial strains and plasmids used in this study.

Strain or Plasmid	Genotype, Description or Use	Reference or Source
** *S. coelicolor* **		
M145	SCP1− SCP2−	[18]
C6S	*scr3559* over-expression: in order to be expressed from the *tipA* promoter, the *scr3559* gene was cloned into the high copy number plasmid pCJW93.	This work
C0	a control strain containing an empty pCJW93 vector	This work
** *E. coli* **		
DH5α	Plasmid construction and general subcloning	Invitrogen
ET12567/pUZ8002	Generation of methylation-free plasmid DNA and conjugation from *E. coli* into *S. coelicolor*	[20]
**Plasmids/cosmids**		
pIJ2925	General cloning vector	[21]
pCJW93	High-copy-number *E. coli/Streptomyces* plasmid containing a thiostrepton inducible (*tipA*) promoter	[22]
StH5	*S. coelicolor* cosmid containing the *scr3559* gene	[23]

**Table 2 microorganisms-09-02004-t002:** Color heat map represents the range of gene expression change in the C6S strain relative to the C0 strain from most down-regulated (dark purple) to most up-regulated (dark maroon) genes.

Gene SCO Number	Average Differential Expression (C6S/C0)	Gene/Cosmid Name	Protein/Gene Cluster
Young Cells (72 h)	Old Cells (144 h)
185	−2.71	3.66	*crtB*/SCJ1.34	putative geranyl pyrophosphate synthase/Isorenieratene
186	−0.15	1.85	*crtE*/SCJ1.35	putative phytoene dehydrogenase (putative secreted protein)/Isorenieratene
187	−0.82	2.53	*crtI*/SCJ1.36	putative phytoene synthase/Isorenieratene
188	−0.41	1.52	*crtV*/SCJ1.37	putative methylesterase/Isorenieratene
189	−0.84	1.14	*crtU*/SCJ1.38c	putative dehydrogenase/Isorenieratene
190	0.02	1.20	*crtT*/SCJ12.02c	Isorenieratene
191	−0.76	1.11	*crtY*/SCJ12.03c	Isorenieratene
216	−1.27	2.72	*narG2*	nitrate reductase alpha chain NarG2
217	−1.61	2.69	*narH2*	nitrate reductase beta chain NarH2
379	1.49	1.05	*katA*/EC 1.11.1.6	catalase KatA
380	0.91	2.10	SCF62.06	conserved hypothetical protein
381	2.37	2.30	SCF62.07	putative glycosyl transferase
382	3.14	0.82	SCF62.08	UDP-glucose/GDP-mannose family dehydrogenase
383	2.24	1.66	SCF62.09	hypothetical protein
384	2.01	1.58	SCF62.10	putative membrane protein
385	1.74	1.23	SCF62.11	putative membrane protein
409	1.77	1.14	*sapA*/SCF51.08c	spore-associated protein precursor
489	−3.72	−6.22	SCF34.08c	conserved hypothetical protein/Siderophore coelichelin
490	−2.18	−3.35	SCF34.09c	putative esterase/Siderophore coelichelin
491	−1.57	−3.33	SCF34.10c	putative ABC transporter transmembrane protein/Siderophore coelichelin
492	−0.88	−3.73	SCF34.11c	putative peptide synthetase/Siderophore coelichelin
493	−1.66	−4.03	SCF34.12c	putative ABC-transporter transmembrane protein/Siderophore coelichelin
494	−2.85	−4.16	SCF34.13c	putative iron-siderophore binding lipoprotein/Siderophore coelichelin
495	−1.32	−1.66	SCF34.14c	putative iron-siderophore ABC-transporter ATP-binding protein/Siderophore coelichelin
496	0.61	−2.29	SCF34.15c	putative iron-siderophore permease transmembrane protein/Siderophore coelichelin
497	−0.40	−2.20	SCF34.16c	putative iron-siderophore permease transmembrane protein/Siderophore coelichelin
498	−0.96	−6.16	SCF34.17c	putative peptide monooxygenase/Siderophore coelichelin
499	−2.28	−4.08	SCF34.18	putative formyltransferase/Siderophore coelichelin
895	−0.57	−0.25	*hrdC*/SCM1.28c	RNA polymerase principal sigma factor HrdC
1849	−2.84	−1.28	*cobN*	cobalamin biosynthesis protein CobN
2077	−1.22	0.55	*divIVA*/SC4A10.10c	DivIVA
2084	−2.10	0.74	*murG*/SC4A10.17c	putative UDP-N-acetylglucosamine-N-acetylmuramyl-(pentapeptide) pyrophosphoryl-undecaprenol N-acetylglucosamine transferase MurG
2431	1.15	−0.88	*abfA*/SCC24.02c	AbfA
2465	0.98	0.31	*hrdA*/SC7A8.04c	RNA polymerase principal sigma factor HrdA
2779	1.86	0.99	*acdH*/SCC105.10	acyl-CoA dehydrogenase AcdH
2780	−1.17	−2.95	SCC105.11	putative secreted protein/Desferrioxamines
2781	−1.12	−1.70	SCC105.12	hypothetical protein/Desferrioxamines
2782	−0.07	−3.94	SCC105.13	putative pyridoxal-dependent decarboxylase/Desferrioxamines
2783	−0.45	−4.20	SCC105.14	putative monooxygenase/Desferrioxamines
2784	−0.61	−4.81	SCC105.15	putative aceytltranferase/Desferrioxamines
2785	−1.29	−4.54	SCC105.16	conserved hypothetical protein/Desferrioxamines
3110	−2.88	−5.29	SCE41.19c	putative ABC transport system integral membrane protein
3111	−4.77	−5.37	SCE41.20c	putative ABC transport system ATP-binding protein
3202	−2.35	−2.27	*hrdD*/SCE22.19c	RNA polymerase principal sigma factor HrdD
3287	0.86	−0.71	SCE15.04	putative serine/arginine rich protein AbeA
3288	7.22	−1.63	SCE15.05	putative integral membrane protein AbeB
3289	5.77	−1.64	SCE15.06	putative large membrane protein AbeC
3290	8.07	−1.80	SCE15.07	hypothetical protein AbeD
3291	0.39	1.53	SCE15.08	putative regulatory protein AbeR SARP
3322	−2.02	1.16	SCE68.20	putative membrane protein, a SCO3558 homolog
3323	3.90	2.51	SCE68.21	putative RNA polymerase sigma factor BldN
3413	3.81	1.28	*tipA*/SCE9.20	transcriptional regulator TipA
3558	−2.38	0.90	SCH5.21	putative morphological differentiation-associated protein
3559	−2.45	−0.08	SCH5.22c	putative oxidoreductase
3607	−2.72	−1.51	SC66T3.18c	putative secreted protein
3608	−3.18	−0.34	SC66T3.19c	hypothetical protein
4654	−1.57	1.74	*rpoB*/SCD82.26	DNA-directed RNA polymerase beta chain RpoB
4655	−1.47	1.11	*rpoC*/SCD82.27	SCDDNA-directed RNA polymerase beta’ chain RpoC
5069	0.40	0.40	SCBAC20F6.12	putative oxidoreductase/Actinorhodin
5070	0.87	0.66	SCBAC20F6.13c	hydroxylacyl-CoA dehydrogenase/Actinorhodin
5071	1.92	2.55	SCBAC20F6.14c	hydroxylacyl-CoA dehydrogenase/Actinorhodin
5072	1.34	1.99	SCBAC20F6.15	hydroxylacyl-CoA dehydrogenase/Actinorhodin
5073	0.74	2.12	SCBAC20F6.16	putative oxidoreductase/Actinorhodin
5074	1.67	1.83	SCBAC20F6.17	putative dehydratase/Actinorhodin
5075	0.92	2.07	*ORF4*/SCBAC28G1.01	putative oxidoreductase/Actinorhodin
5076	0.67	0.24	*actVA1*/SCBAC28G1	integral membrane protein/Actinorhodin
5077	−0.09	1.19	*actVA2*/SCBAC28G1	hypothetical protein/Actinorhodin
5078	1.65	0.86	*actVA3*/SCBAC28G1	hypothetical protein/Actinorhodin
5079	2.62	0.31	*actVA4*/SCBAC28G1	conserved hypothetical protein/Actinorhodin
5080	1.25	−0.41	*actVA5*/SCBAC28G1	putative hydrolase/Actinorhodin
5081	0.48	2.40	*actVA6*/SCBAC28G1	hypothetical protein/Actinorhodin
5082	−1.18	0.42	*actII-1*/SCBAC28G1	putative transcriptional regulatory protein/Actinorhodin
5083	−0.12	−2.64	*actII-2*/SCBAC28G1	putative actinorhodin transporter/Actinorhodin
5084	1.31	−3.45	*actII-3*/SCBAC28G1	putative membrane protein/Actinorhodin
5085	−0.82	5.01	*actII-4*/SCBAC28G1	actinorhodin cluster activator protein/Actinorhodin
5086	0.64	2.27	*actIII*/SCBAC28G1	ketoacyl reductase/Actinorhodin
5087	0.76	1.90	*actI-1*/SCBAC28G	actinorhodin polyketide beta-ketoacyl synthaselpha subunit/Actinorhodin
5088	0.90	0.84	*actI-2*/SCBAC28G	actinorhodin polyketide beta-ketoacyl synthaseeta subunit/Actinorhodin
5089	0.67	2.72	*actI-3*/SCBAC28G	actinorhodin polyketide synthase acyl carrierprotein/Actinorhodin
5090	0.02	2.27	*actVII*/SCBAC28G1	actinorhodin polyketide synthase bifunctionalyclase/dehydratase/Actinorhodin
5091	−0.07	1.54	*actIV*/SCBAC28G1.1	cyclase/Actinorhodin
5092	0.61	0.80	*actVB*/SCBAC28G1.1	actinorhodin polyketide putative dimerase/Actinorhodin
5572	−3.99	2.24	*rnc*/SC7A1.16	ribonuclease III/RNase III
5621	−1.70	3.75	*whiG*/SC2E1.38	RNA polymerase sigma factor WhiG
5819	1.33	1.96	*whiH*	Sporulation transcription factor WhiH
5820	−0.49	1.97	*hrdB*/SC5B8.10	major vegetative sigma factor HrdB
5877	1.96	3.21	*redD*	transcriptional regulator RedD/Prodiginines
5878	1.34	0.94	*redX*	polyketide synthase RedX/Prodiginines
5879	0.33	0.32	*redW*	acyl-coa dehydrogenase RedW/Prodiginines
5880	0.09	3.34	*redY*	RedY protein/Prodiginines
5881	−1.54	2.78	*redZ*	response regulator/Prodiginines
5882	0.61	0.02	*redV*/SC3F7.02c	Prodiginines
5883	0.79	0.87	*redU*/SC3F7.03c	Prodiginines
5884	1.41	0.09	SC3F7.04c	hypothetical protein/Prodiginines
5885	1.21	−0.66	SC3F7.05c	hypothetical protein/Prodiginines
5886	1.68	2.31	*redR*/SC3F7.06c	Prodiginines
5887	2.60	0.20	*redQ*/SC3F7.07c	Prodiginines
5888	0.94	−0.50	*redP*/SC3F7.08	3-oxoacyl-[acyl-carrier-protein] synthase/Prodiginines
5889	2.70	0.50	*redO*/SC3F7.09	hypothetical protein/Prodiginines
5890	1.65	0.89	*redN*/SC3F7.10	putative 8-amino-7-oxononanoate synthase/Prodiginines
5891	1.02	−0.79	*redM*/St3F7.11	putative peptide synthase/Prodiginines
5892	0.92	−0.28	*redL*/SC3F7.12	polyketide synthase/Prodiginines
5893	0.82	0.08	*redK*/SC3F7.13	oxidoreductase/Prodiginines
5894	0.69	0.02	*redJ*/SC3F7.14	thioesterase/Prodiginines
5895	1.24	−0.04	*redI*/SC3F7.15	putative methyltransferase/Prodiginines
5896	1.45	0.17	SC10A5.01	phosphoenolpyruvate-utilizing enzyme/Prodiginines
5897	2.47	0.76	*redG*/SC10A5.02	Prodiginines
5898	2.38	1.57	*redF*/SC10A5.03	Prodiginines
6265	−0.25	0.51	*scbR*/SCAH10.30c	Gamma-butyrolactone binding protein
6266	1.63	1.14	*scbA*/SCAH10.31	SCB1 butanolide synthase, an AfsA homolog/Gamma-butyrolactone
6267	−0.44	1.19	SCAH10.32	hypothetical protein, a BprA homolog
6992	1.82	5.48	*absR1*/SC8F11.18c	regulatory protein AbsR1
6993	1.31	2.10	*absR2*/SC8F11.19	regulatory protein AbsR2
7221	−0.75	3.04	SC2H12.20c	Gcs type III polyketide synthase

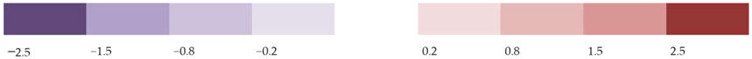

## Data Availability

Not applicable.

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
