# Peer review of "6S-Like scr3559 RNA Affects Development and Antibiotic Production in Streptomyces coelicolor"

_microorganisms, 2021, doi:10.3390/microorganisms9102004_

Round 1
Reviewer 1 Report
Bobek and coworkers have addressed all of the concerns that were raised. Moreover, they have made a few interesting additions to their initial submission. I feel the manuscript has much improved and is now suitable for publication.
Minor issues:
The new panel B that was added to Fig. 2 has a lot of white space in it, while the details of the RNA structures are too small to see. It would be good to enlarge and reorient the three RNAs to better fill the page. It could also help to indicate the 3'- and 5'-ends of the molecules, which (especially for the mature scr3559) are very difficult to see.
L. 319-320: "a cluster of genes encoding for methylenomycin furans (MMFs)" - presumably the authors mean to say that this cluster encodes enzymes responsible for the synthesis of MMFs?
Author Response
Dear reviewer,
thank you for your extensive revision of our manuscript!
- The secondary structures in Figure 2b have been enlarged and rearranged to minimize the white space around. The 5’ and 3’ ends of the molecules have been highlighted and a sentence describing these indications has been added to the figure legend (lines 184-5).
- Line 322: As you suggested, the following text has been added in discussion: ”...enzymes responsible for the synthesis of...“.
Reviewer 2 Report
The authors responded fully to the previous comments. The figures are clear and, some points have been introduced in the text. Therefore, I recommend accepting the manuscript in this form.
Author Response
Dear reviewer!
Thank you again for all your suggestions that highly improved our manuscript!
This manuscript is a resubmission of an earlier submission. The following is a list of the peer review reports and author responses from that submission.
Round 1
Reviewer 1 Report
In this manuscript, the authors study the effect of the overexpression 6S-like RNA transcript (scr3559) in S. coelicolor. 6S RNA affects both development and actinorhodin and undecylprodigiosin production in S. coelicolor. The authors firstly use 5’-RACE to describe the pre-RNA and the mature form of 6S RNA, after this verify the expression of 6S-RNA in WT and C6S strains using Northern Blot. Finally, the authors measure actinorhodin, undecylprodigiosin and other antibiotics (LC-MS). Moreover, the authors measure changes in expression using microarray analysis.
This manuscript is well written, the experiments are well designed, and I enjoyed reading it. I have only a few minor comments.
The first point is the 5’sequence of pre-scr3559 RNA that is removed. The data of RT-PCR and 5’-RACE are very significant and newly. However, these data raise some questions. Is there any biological or computational evidence that suggests the significance of this maturation process? What is a hypothetical function of the sequence in 5’ of the pre-scr3559 RNA? It is possible that this sequence masks the nucleotides needed to bind the RNAP-HrdB? These are very interesting points. I suggest mapping the position of the 68 nt removed in a 3D or 2D structure of the RNA 6S. Otherwise, I suggest modelling 3D or 2D structures of pre-RNA and mature RNA. These methods can respond to some questions.
Comments
Thiostrepton is an antibiotic that binds ribosomes and can influence the stringent response. The authors designed the experiment correctly (adding control). I suggest adding values (from literature) of MIC and MBC for using thiostrepton on Streptomyces cultures.
Fig. 1. Add RNA 6S as a tag for scr3559.
Fig. 2. The 192 nt band is difficult to see. Provide a figure with better contrast or enlarge the figure. The words "3X3" and "4" on the figure could be moved to the right (near the arrows).
Fig. 3. C6S 48 h. There are two bands. Describe these bands in legend or at the right of the image.
Fig. 4b and 4c. please insert the name of the y-axis
Table 2. The common names of genes in Italic
Some very recent studies on ncRNAs of Streptomyces can be added in the introduction or discussion.
Reviewer 2 Report
Bobek et al. describe the effects of overexpressing a known regulatory RNA molecule, scr3559, in Streptomyces coelicor. A knock-out of the corresponding gene had already been analysed in some detail in earlier work, demonstrating that scr3559 influences the production of the antibiotic actinorhodin. In their present manuscript, the authors report that plasmid-mediated overexpression of scr3559 leads to upregulation of antibiotic production as well as morphological changes. In the light of the previous study, this outcome seems credible and maybe even somewhat predictable. The manuscript is clearly written and well-structured.
The main problem I have with this work is its complete lack of rigorous statistics. Although the authors regularly (l. 55, 197, 234, 237) use the word "significant" in a colloquial manner - which, on a side note, is best avoided in scientific texts - the actual statistical significance of many of the outcomes seems doubtful to say the least.
For instance, the authors do not indicate what the error bars in Fig. 4 denote (standard deviation, standard error in the mean, something else?), nor do they mention the number of experimental repeats underlying those values. I also feel that some kind of explanation is in order as to why the error bars are tiny for some of the experiments and huge for others. But most importantly: do the authors truly believe that the differences between C0 and C6S (particularly at 144 h, left panel) are significant and, if so, can they argue why (for instance, by using a statistical test)? Why is the 144 h time point shown for actinorhodin, but not for undecylprodigiosin?
Similar issues exist with the LC-MS-based analysis of various secondary metabolites. As far as I can see, the values that are reported are outcomes of single experiments, without any repeats or error estimates. Given that (as Fig. 4 shows) variation in the S. coelicor cultures can be very large indeed, how do the authors know that the differences that they observe in these assays are statistically meaningful?
The same criticism applies to the microarray-based comparison of gene transcription levels. Moreover, the time points that are chosen (72 h and 144 h) appear to correspond to maximum accumulation of the scr3559 precursor molecule, while the processed RNA (the "functional form" according to l. 163, although this notion is not further developed anywhere in the text) has already completely disappeared at this stage (Northern blot, Fig. 3). Consequently, what causes the transcriptional changes remains unclear: the presence of the precursor RNA, the former presence of the processed RNA, a combination of these, or maybe indirect effects related to differences in development? I am also wondering why the authors chose not to analyse the 48 h time point, when the mature/functional scr3559 molecule is at its most abundant according to Fig. 3?
Fig. 6: the right-hand panel (repeat of the experiment but this time on ONA medium) is not mentioned anywhere in the main text. Moreover, the result in this panel appears to contradict the statement that the C0 control strain "did not show any phenotypic differences on the ... thiostrepton gradient" (l. 216), requiring an explanation.
Minor issues:
Fig. 5: the scale bars on the electron micrographs are difficult to make out (black-and-white stripes on a black background). Bar colour should be changed.
Fig. 6: if the dotted line indeed indicates zero thiostrepton (i.e. the end of the gradient) as mentioned in the legend, the blue gradient triangles on the right-hand side should not extend all the way down to the bottom of the plate.
The manuscript should be carefully checked for minor language issues and typos (e.g. "boarder", l. 220)